# Key Processing Factors in Hydrothermal Liquefaction and Their Impacts on Corrosion of Reactor Alloys

Minkang Liu and Yimin Zeng *

Natural Resources Canada, CanmetMATERIALS, Hamilton, ON L8P 0A5, Canada;
minkang.liu@nrcan-rncan.gc.ca
* Correspondence: yimin.zeng@nrcan-rncan.gc.ca

**Abstract:** Despite intensive efforts to develop hydrothermal liquefaction for the conversion of wet biomass and biowaste feedstocks into valuable bio-oils, severe corrosion of conversion reactor alloys and other core components, induced by the pressurized hot water medium, catalysts, and inorganic and organic corrodants generated during the conversion process, has significantly hindered the industrial deployment of this attractive technology. In this paper, a general review of major operating parameters, including biomass feedstock types, temperature, pressure, and catalysts, was conducted to advance the understanding of their roles in conversion efficiency and the yield and properties of produced oils. Additionally, the corrosion performance of a representative constructional alloy (Alloy 33) was investigated in both non-catalytic and catalytic HTL environments at temperatures of 310 °C and 365 °C, respectively. The alloy experienced general oxidation in the non-catalytic HTL environment but suffered accelerated corrosion (up to 4.2 μm/year) with the addition of 0.5 M $K_2CO_3$ catalyst. The corrosion rate of the alloy noticeably increased with temperature and the presence of inorganic corrodants ($S^{2-}$ and $Cl^-$) released from biowastes. SEM/XRD characterization showed that a thin and compact Cr-rich oxide layer grew on the alloy in the non-catalytic HTL environment, while the surface scale became a double-layer structure, composed of outer porous Fe/Cr/Ni oxides and inner Cr-rich oxide, with the introduction of the $K_2CO_3$ catalyst. From the corrosion perspective, the alloy is a suitable candidate for construction in the next phase of pilot-scale validation assessment.

**Keywords:** hydrothermal liquefaction; major operating factors; corrosion; $K_2CO_3$ catalyst; inorganic corrodants





## 1. Introduction

In 2020, $CO_2$ emissions accounted for approximately 79% of total U.S. anthropogenic greenhouse gas (GHG) emissions alone [1]. Among the sources of $CO_2$ emissions, fossil fuel combustion for energy accounted for 73% of total U.S. GHG emissions and a staggering 92% of total U.S. anthropogenic $CO_2$ emissions [1]. Tremendous research and development (R&D) efforts have been dedicated to exploring sustainable and environmentally friendly alternative energy resources. Among these, biomass stands out as the world's fourth-largest energy source, holding great potential due to its renewable nature and significant benefits for local and international economic growth. Recent estimates suggest that liquid biofuels derived from diverse biomass feedstocks could potentially contribute up to approximately 15% of global energy consumption [2]. By harnessing the power of biomass and other clean and renewable energy sources, including wind and solar, it is anticipated that more than one-third of the world's primary energy needs in the near future could be fulfilled [3–5].

However, raw biomass feedstocks alone cannot be rapidly converted into bioenergy products at the scale necessary to meet the global annual energy demand, which was approximately 226 exajoules in 2022, according to a recent study [6]. To bridge this gap, innovative conversion technologies are essential in transforming biomass into usable bioenergy products more efficiently and effectively. To date, the developed conversion

technologies are classified into two main categories: biochemical and thermochemical methods [7]. Compared to biochemical conversion, thermochemical conversions involve processes conducted at elevated temperatures and with the assistance of suitable catalysts. These pathways enable the production of a wide range of value-added bioenergy products, including syngas from gasification, solid biochar from torrefaction, and bio-oils from fast pyrolysis and hydrothermal liquefaction. These products generally possess higher heating value (HHV) and lower moisture content, making them highly desirable for energy applications [8]. The residence time of thermochemical conversions is typically shorter compared to biochemical processes, leading to higher conversion efficiencies. Thermochemical conversions are further classified into various methods, including pyrolysis, hydrothermal liquefaction, gasification, torrefaction, and combustion, based on the biomass feedstocks, operating conditions, and desired final products [9]. Among the different thermochemical pathways, hydrothermal liquefaction (HTL) stands out as a highly promising process for the production of biofuels with high heating value. HTL has the advantage of directly converting wet biomass and biowastes, eliminating the need for energy-intensive drying processes required by other conversion technologies. This not only improves the overall energy efficiency but also offers cost-effective biowaste management solutions. In a HTL process, raw biomass is mixed with a solvent, typically water, and heated to temperatures ranging from 250 to 374 °C, and pressures up to 25 MPa [10,11]. Under such conditions, the biomass undergoes a series of reactions, primarily depolymerization and decomposition, which involve the breakdown of complex organic compounds and the recombination of the produced intermediates. This process ultimately transforms the biomass into bio-oil, also known as biocrude, along with other byproducts such as gases and solid residues. The produced bio-oil can be further upgraded to drop-in fuel or other desired biochemicals.

Despite significant advancements in hydrothermal liquefaction (HTL) technology, its industrial application has yet to be established due to challenges, such as high capital investment, clogging, and potential corrosion of key components, including the conversion reactor and core components. Therefore, this study aimed to advance fundamental understanding of key operating parameters on HTL processes, and to investigate the influence of these factors on corrosion of candidate construction alloys to fill some knowledge gaps on the cost-effective construction and long-term safe operation of HTL plants. In this study, the influence of temperature, $K_2CO_3$ catalyst, and corrodants presented in biowastes on the corrosion of a representative constructional alloy (Alloy 33) were investigated in simulated HTL environments using a standard high temperature autoclave testing methodology and advanced characterization techniques.

### 1.1. Overview of Hydrothermal Liquefaction (HTL) Process and Candidate Construction Alloys

The HTL process encompasses a series of chemical reactions, including hydrolysis, depolymerization, dehydration, repolymerization, and condensation [12]. During hydrolysis, water molecules break down complex organic compounds into simpler constituents through the cleavage of chemical bonds. Depolymerization occurs as the resulting molecules undergo further fragmentation, leading to the breakdown of large polymers into smaller molecules. Dehydration involves the removal of water molecules from these smaller compounds, leading to the formation of reactive intermediates. These intermediates can then undergo repolymerization and condensation reactions, where the smaller molecules recombine to form larger and more stable compounds, such as bio-oils or solid char. During HTL conversion, forest and agricultural feedstocks undergo direct disintegration, resulting in the formation of cellulose, hemicellulose, and lignin fragments. These fragments are then converted into valuable bioenergy products. Additionally, various biowaste streams, including sewage sludge, black liquor, hog fuel, and plastic-contaminated streams, can also be transformed into bioenergy products through de-polymerization, decarboxylation, and aromatization reactions within the HTL conversion process [13,14]. Numerous studies [15,16] have also demonstrated that the bio-oil produced through HTL exhibits significantly lower oxygen content compared to oil generated from fast pyrolysis. This

characteristic makes HTL bio-oil highly suitable for the production of drop-in fuels through direct upgrading or co-processing technology at existing petroleum plants. Based on our literature review, the properties of HTL bioproducts are primarily influenced by three factors: the types of biomass feedstocks, the operating temperature and pressure, and the catalysts used. Consequently, the impacts of these factors on HTL conversion efficiency are reviewed in the subsequent sections. Additionally, the corrosion of core component alloys is widely recognized as one of the key technical challenges in the commercialization of HTL technology [10,17]. The corrosion behaviors of candidate constructional materials are also thoroughly reviewed and discussed, specifically focusing on their performances in existing high-temperature, high-pressure systems.

*1.2. Biomass Feedstocks*

Biomass is typically composed of a complex mixture of carbohydrates, proteins, and minerals, which gives it unique chemical and physical characteristics [18,19]. Depending on the source of origin, biomass feedstocks used in HTL conversion can be classified into categories such as agricultural and forest biomass, specially engineered algae, and domestic municipal and industrial waste streams. The utilization of agricultural and forest biomass feedstocks, such as straw, corn stalks, and both soft and hard woods, has several merits [20]. These traditional biosources offer high yield potentials and benefit from existing supply chains. However, their usage can lead to competition with food production, conflicts regarding land use, and limited resource efficiency. To address these drawbacks, specially engineered biological feedstocks have been developed, consisting of microalgae or macroalgae. These engineered feedstocks offer advantages, such as high productivity, continuous harvesting, and high lipid contents [21]. Additionally, they have the potential to facilitate carbon dioxide fixation and wastewater treatment. However, this generation of feedstocks also faces limitations, including challenges in scaling up production, nutrient requirements, and difficulties in harvesting and dewatering. Another promising feedstock for hydrothermal liquefaction is municipal and industrial waste streams. These waste streams are the most complex and typically consist of highly aggressive components, like hydroxide, sulfide, and thiosulphate. The conversion of these waste streams into valuable bioproducts offers significant advantages, such as waste valorization, reduction of landfill usage, and the potential for decentralized processing. However, these waste streams also present certain drawbacks. Their compositions and qualities can vary, and they may contain contaminants. Therefore, pre-processing is often necessary to remove impurities and optimize conversion processes.

Table 1 presents the biochemical compositions of representative biomass feedstocks. Forest and agricultural biomasses/residues predominantly consist of three compounds: cellulose, hemicellulose, and lignin [22]. It is important to note that swine manure, listed in the table, is also considered an agricultural residue, containing certain amounts of protein and lipid. In contrast, algae feedstocks typically contain protein, carbohydrate, and lipid. Biowastes, on the other hand, generally have lower protein content, while their carbohydrate and lipid contents vary depending on the source. Elemental analyses reveal that forest and agricultural biomasses tend to have higher oxygen content, while carbon and hydrogen contents do not exhibit a clear trend. It is worth noting that the components and elements within biowastes can significantly differ, depending on their sources and processing conditions. Despite the varying compositions of biomass feedstocks, the general conversion mechanism under HTL conditions involves the breakdown of intermolecular hydrogen bonds into monomers, followed by the hydrolysis of biopolymers. However, the chemical content (O, H, C), structural components (hemicellulose, lignin), lipid, and carbohydrate composition can significantly differ among different biomass feedstocks. This variation affects the reaction processes, kinetics, and, ultimately, the yield and quality of biocrudes. Recent studies [23,24] have attempted to predict the yield and quality of biocrude from different feedstocks through kinetic modeling and laboratory-scale testing. However, a comprehensive global kinetic model for predicting HTL product yields, based

on feedstock chemistry and processing conditions, is still lacking. Table 2 summarizes the final bioproducts obtained from different feedstocks after HTL conversion at temperatures of 300–350 °C for 30–60 min, based on previous studies [23–28]. Waste streams and microalgae tend to yield significantly higher biocrude compared to agricultural biomass, with food waste showing the highest yield at 38.8 wt.%. This can be attributed to its higher carbohydrate content. Additionally, food waste feedstocks exhibit the lowest oxygen content in the final products. These findings suggest that HTL could be a more suitable method for processing waste streams and specially engineered algae compared to other thermochemical technologies. To summarize, the composition of biomass feedstocks significantly affects the HTL conversion mechanism, yield, and quality of biocrudes. Recent studies have examined biocrude production and quality prediction, revealing a need for a comprehensive global kinetic model. HTL proves effective in converting waste streams and engineered microalgae, making it a favorable choice among thermochemical technologies.

**Table 1.** Biochemical compositions of representative HTL feedstocks.

| | Softwood [25,29,30] | Hardwood [25,29,31] | Corn Stalk [27,28] | Swine Manure [24] | Spirulina Algae [26] | Food Waste [23] | Sewage Sludge [26] |
|---|---|---|---|---|---|---|---|
| Lipid (wt.%) | – | – | – | 8.80 | 7.50 | 21.9 | 6.90 |
| Protein (wt.%) | – | – | – | 21.9 | 63.4 | 17.8 | 33.6 |
| Carbohydrate (wt.%) | – | – | – | – | 20.5 | 59.1 | 20.3 |
| Cellulose (wt.%) | 43.2 | 39.4 | 32.8 | 10.1 | – | – | – |
| Hemicellulose (wt.%) | 28.4 | 22.3 | 26.5 | 28.2 | – | – | – |
| Lignin (wt.%) | 13.8 | 19.5 | 12.6 | 4.46 | – | – | – |
| C (wt.%) | 51.0 | 48.6 | 54.9 | 41.2 | 43.2 | 58.3 | 43.4 |
| H (wt.%) | 6.2 | 6.0 | 4.8 | 5.67 | 8.53 | 10.3 | 5.91 |
| N (wt.%) | 0.74 | 0.85 | 2.54 | 3.86 | 8.91 | 2.00 | 3.18 |
| O (wt.%) | 43.5 | 45.4 | 25.3 | 32.5 | 39.4 | 29.3 | 47.5 |
| Ash (wt.%) | 0.2 | 0.6 | 13.2 | 17.0 | 8.60 | 1.10 | 39.2 |
| HHV (MJ/Kg) | 18.9 | 17.6 | 15.6 | – | 19.8 | 24.6 | 14.6 |

**Table 2.** Biocrude yield and compositions of representative bioproducts after HTL conversion.

| | Softwood [25,29,30] | Hardwood [25,29,31] | Corn Stalk [27,28] | Swine Manure [24] | Spirulina Algae [26] | Food Waste [23] | Sewage Sludge [26] |
|---|---|---|---|---|---|---|---|
| Biocrude yield (wt.%) | 16.30 | 15.80 | 19.70 | 25.60 | 39.50 | 38.80 | 34.50 |
| C (wt.%) | 75.20 | 74.60 | 68.40 | 75.60 | 72.50 | 79.00 | 70.30 |
| H (wt.%) | 8.60 | 8.50 | 9.58 | 8.96 | 9.44 | 10.30 | 8.72 |
| N (wt.%) | 0.85 | 1.11 | 1.60 | 4.54 | 6.96 | 4.40 | 9.70 |
| O (wt.%) | 14.20 | 14.50 | 17.80 | 10.90 | 11.10 | 6.30 | 11.30 |
| Energy (MJ/kg) | 36.20 | 35.00 | 35.60 | 36.30 | 36.10 | 35.60 | 34.30 |

### 1.3. Operating Parameters

Apart from biomass feedstocks, the operating temperature and pressure are recognized as critical factors in reaction kinetics and biocrude yields. Increasing pressure generally promotes hydrolysis reactions and enhances depolymerization to some extent [35]. However, temperature plays a more dominant role in the conversion process, affecting factors such as complete biomass conversion, overall yield of bioproducts, and the physicochemical properties of the final products [36]. It is worth noting that these two parameters also impact the cost of the conversion process and the long-term structural integrity of core components in HTL biorefinery plants, as highlighted in recent publications [10,37].

Based on previous studies, HTL conversion, conducted within the temperature range of 200–400 °C, is categorized and summarized in Table 3. It should be noted that alkaline catalysts, such as 0.5–1.0 M $Na_2CO_3$ and $K_2CO_3$, were commonly employed in these studies. As observed in the table, the biocrude yield tends to increase with temperature, with an optimized temperature range potentially lying between 300 and 380 °C [38,39]. In the case of forest and agricultural feedstocks, raising the temperature significantly improves the hydrolysis rates of cellulose, hemicellulose, and lignin. Additionally, higher temperatures could potentially accelerate the fragmentation and degradation of microalgae, as depicted in Table 3. Therefore, the primary role of increasing temperature is to surpass the transition energy barrier for raw biomass and its corresponding intermediate products, aiding in the breakdown of chemical bonds. For instance, a previous study [40] investigated the HTL of barley straw within the temperature range of 280–400 °C, utilizing a 0.75 M $K_2CO_3$ catalyst. The yield of biocrude oil consistently increased from 280 to 350 °C, reaching up to 34.9 wt.%.

In addition to the aforementioned advantages, increasing temperature in HTL has the potential to reduce the oxygen content in the produced organic phases, which is beneficial for subsequent direct upgrading or co-processing with petroleum intermediates. For instance, a study investigating the hydrothermal liquefaction of hardwood within the temperature range of 280 to 360 °C observed a simultaneous decrease in oxygen content as temperature increased [41]. However, the effect of increasing temperature on the heating value of the generated biocrudes is marginal. This is likely due to the fact that changes in hydrogen and carbon content within the organic products, influenced by temperature, have a combined effect that tends to balance out their impact on the HHV [42]. It is worth noting that the biocrude yields and properties are also influenced by various factors, such as the types of solvents, catalysts, residence time, water–biomass ratio, and heating rate during the HTL conversion, which should be taken into consideration.

**Table 3.** Effect of temperature on HTL yields and physicochemical properties of final products.

| Feedstock | Operating Temperature (°C) | Biocrude Yield (wt.%) | HHV (MJ/kg) | C (wt.%) | O (wt.%) | Ref. |
|---|---|---|---|---|---|---|
| Softwood | 280–360 | 19.3–23.8 | 27.1–30.2 | 69.6–75.1 | 19.2–24.8 | [43] |
| Softwood | | 13.8–25.8 | 28.3–31.9 | 75.4 | 18.8 | |
| Softwood | 277–377 | 27.9–31.6 | 75.2 | 6.1 | - | [44] |
| Hardwood | | 27.6–31.3 | 75.1 | 6.0 | - | |
| Hardwood | 280–360 | 18.6–27 | 26.2 | 66.8 | 27.3 | [45] |
| Hardwood | 250–400 | 15–28.4 | 17.4–34.5 | 63.8–77.9 | 14–34.7 | [41] |
| Corn stover | 250–375 | 14.3–27.2 | 27.5–35.1 | 64.9–76.3 | 7.2–8.2 | [36] |
| Barley straw | 280–400 | 19.9–34.9 | 26.8–35.5 | - | - | [40] |
| Cherry stones | 200–300 | 3–6 | 20.8–22.5 | 58 | 34–36 | [46] |
| Spirulina algae | 200–380 | 18–39.9 | 25.2–39.9 | 55.5–82.1 | 0.6–28.9 | [47] |

Although the operating pressure may not have a pronounced influence compared to temperature, it can still affect the kinetics of HTL conversion and the final biocrude yield, as discussed earlier. The change in pressure can promote or hinder hydrolysis reactions and depolymerization during the process [35]. In the subcritical region, increasing pressure leads to an increase in the density of the solvent. For example, at 330 °C, increasing pressure from 25 to 35 MPa results in an increase in water density to 22 kg/m$^3$, based on the NIST chemistry database. This higher solvent density allows for better penetration and interaction of hydrolysis ions within the crystallized zone of cellulose and the surface of cellulose, hemicellulose, and lignin. A study on the HTL of softwood [41] found that increasing pressure led to an increase in biocrude yield from 18 to about 25% at 350 °C.

However, the role of pressure may change within the supercritical zone. For example, Chan et al. [48] studied the HTL of agricultural biomass (fruit bunch and palm shell) and found that the biocrude yield decreased by 6–9 wt.% in the final product when the operating pressure increased from 25 to 30 MPa at 390 °C. As shown, some previous studies suggest that elevated pressure levels within the subcritical temperature range may have a favorable impact on biocrude production. However, a general conclusion regarding the impact of pressure on biocrude yield has not yet been achieved. In a nutshell, conversion temperature and pressure are critical factors influencing reaction kinetics and biocrude yields in HTL. Increasing temperature promotes hydrolysis rates and fragmentation of biomass, leading to higher biocrude yields and reduced oxygen content. Pressure, while being less influential, can enhance depolymerization kinetics, leading to better penetration and interaction of hydrolysis ions, resulting in higher biocrude yields. While some studies suggest a favorable impact of elevated pressure in the subcritical range, a conclusive understanding of pressure's overall effect on biocrude yield is yet to be achieved.

### 1.4. Catalysts

In addition to the operating parameters, the catalyst applied is a critical factor that affects the physicochemical properties and yield of biocrude during HTL conversion. Both homogeneous and heterogeneous catalysts have been extensively studied for this purpose [49,50]. While heterogeneous catalysis has traditionally been associated with gasification processes, several studies have investigated the efficacy of various heterogeneous catalysts, such as Fe, Pd, Ni, Pt, and $Al_2O_3$, in the HTL conversion process. It has been found that the addition of these catalysts can promote deoxygenation and increase the HHV and quality of biocrude oil [51–53]. For example, a previous study [51] reported that the application of heterogeneous catalysts, such as carbon nanotubes and Pt/C, increased the HHV of biocrude oil produced from microalgae by 10%. Another study [54] found that using Fe metal as a catalyst resulted in biocrude with the highest carbon and hydrogen contents, as well as HHV, when converting lignocellulosic biomass.

While heterogeneous catalysts have their advantages, such as increased HHV and improved biocrude quality, they can be costly and may have limited activity and accessibility during HTL conversions. Clogging can also be a potential concern when using heterogeneous catalysts in pilot-scale continuous reactor systems [55]. As a result, homogeneous alkaline catalysts, such as $Na_2CO_3$, K2CO3, NaOH, and KOH, have been extensively studied in recent years. These catalysts have been found to significantly increase biocrude yield when introduced into the solvent during HTL conversion [56]. Table 4 provides a summary of the influence of homogeneous catalysts on the efficiency of HTL conversion and their optimum concentrations. For example, one study [57] suggested that the addition of 1 M KOH or $K_2CO_3$ in the HTL solvent increased biocrude oil yield from pinewood by 26% under the same conversion conditions, while also reducing the oxygen content in the final products. The introduction of alkaline catalysts can have a significant effect on the liquefaction behavior of proteins and carbohydrates present in waste streams and algae feedstocks, promoting hydrolysis reactions [58].

In general, the addition of 0.5–1.0 M of an alkaline catalyst can greatly enhance both the conversion efficiency and product quality during biomass HTL conversion. The overall rank of catalyst activity in HTL conversion has been suggested as $K_2CO_3$ > KOH > $Na_2CO_3$ > NaOH. However, it is important to note that using homogeneous catalysts has drawbacks. One major challenge is isolating the product from the liquid media, and another concern is the accelerated corrosion of reactor alloys, due to changes in the environmental chemistry of the process environmental. For example, the addition of 0.5 M $K_2CO_3$ can increase the environmental pH to approximately 11.2, causing major alloying elements (Cr, Fe, Ni) to suffer active corrosion, instead of passivation, during the conversion process [17].

**Table 4.** Different concentrations of catalysts used in HTL conversions.

| Biomass Feedstocks | Catalyst Used | Major Observations | Ref. |
|---|---|---|---|
| Softwood | 0, 0.25, 0.5 and 1 M $K_2CO_3$ | 0.5 M to 1 M gives the highest bio-oil yield. | [59] |
| Softwood | 0.235, 0.47 and 0.9 M $K_2CO_3$ | 0.94 M is the best concentration to produce high oil yield and less residues. | [60] |
| Softwood | 1 M NaOH, $Na_2CO_3$, KOH and $K_2CO_3$ | Yield of bio-oil increases from 8 wt.% up to 34 wt.%. Catalytic activity rank: $K_2CO_3$ > KOH > $Na_2CO_3$ > NaOH. | [57] |
| Corn stalk | 0.1 M $Na_2CO_3$ | Increases the yield of liquid product and its quality, the biocrude yield also increases from 33.4 wt. % to 47.2 wt.% | [61] |
| Palm fruit bunch | 0.1–2.0 M $K_2CO_3$ | 1 M $K_2CO_3$ gives the best conversion efficiency | [62] |
| Microalgae | 0.4 M $Na_2CO_3$ | Highest oil yield of 25.8% achieved with addition of 0.4 M $Na_2CO_3$. | [63] |
| Manure digestate & food waste | 0.1 M NaOH | Manure digestate: increase yield of biocrude oil by 15%, increase yield of aqueous products by 39% and decreases hydro-char. Food waste: lower yield of biocrude oil, lower aqueous product, higher hydro char. | [56] |
| Swine carcasses | 0.2–0.75M NaOH | Yield of biocrude oil increases from 4.5 wt.% to 23.5 wt.%. | [64] |
| Wet organic waste streams | 0.75 M $K_2CO_3$, 1.5 M $K_2CO_3$ | When catalyst was reduced from 1.5 M to 0.75 M, the amount of the oil phase reduced by 2 wt.%. | [65] |

To put it concisely, in the HTL conversion process, the choice of catalyst is crucial in influencing the yield and quality of biocrude. Heterogeneous catalysts, such as Fe, Pd, Ni, Pt, and $Al_2O_3$, have shown promise in promoting deoxygenation and enhancing the HHV of biocrude. However, the cost, limited activity, and potential clogging issues associated with heterogeneous catalysts have led to increased research on homogeneous alkaline catalysts, such as $Na_2CO_3$, $K_2CO_3$, NaOH, and KOH, which have been found to significantly increase biocrude yield and reduce oxygen content. The addition of 0.5–1.0 M alkaline catalysts has shown potential to improve conversion efficiency and product quality, although challenges in isolating the product and dealing with corrosion issues persist.

*1.5. Corrosion Concern and Candidate Materials Selection for Core Components Construction*

As mentioned earlier, corrosion poses a significant challenge to the commercialization of HTL technology. High-temperature aqueous corrosion is a primary concern in HTL reactors, as they involve the use of high-temperature, high-pressure water as the conversion medium. Our recent findings indicate that operating pressure has only a marginal effect on corrosion [10]. However, the addition of homogeneous alkaline catalysts can significantly impact the corrosion modes and extent of construction alloys, as most Fe, Cr, and Ni oxides are thermodynamically unstable in aqueous environments with a pH higher than 11 [17]. Furthermore, biowastes contain substantial amounts of chloride and sulfide ions, which are released during the conversion process and can promote various forms of corrosion, including pitting and stress corrosion cracking, in Fe-based and Ni-based alloys [66,67]. To address these corrosion challenges, it is crucial to select appropriate construction alloys, optimize process parameters, and develop cost-effective corrosion-resistant coatings. While commercial steels and alloys, such as austenitic, ferritic-martensitic, duplex stainless steels, nickel-based alloys, titanium alloys, and zirconium alloys, have been widely used in the high-temperature systems of boiler tubes, autoclave liners, and nuclear reactor cores, and similar, not all of them are suitable for constructing industrial-scale HTL plants, due to economic and practical considerations [10,17]. For instance, zirconium and titanium alloys exhibit excellent corrosion resistance in highly aggressive environments but are not practical for construction due to their high costs. Among the available materials, austenitic stainless

steels and nickel-based alloys, based on the performance and knowledge gained from pressurized water nuclear reactors, are the primary candidates for HTL reactor construction. These materials demonstrate promising high-temperature mechanical properties and acceptable corrosion resistance in high-temperature environments [68,69]. Additionally, our previous studies in high-temperature supercritical water environments suggest that high chromium-based alloys could also be suitable candidates [70,71]. However, limited research has been conducted on corrosion in biomass thermochemical conversion environments to date. Among all the candidate construction materials, Alloy 33 has been considered a promising candidate due to its application in similar high-temperature aqueous systems. Based on a previous study [72], Alloy 33, a chromium-based austenitic alloy, was found to develop a thin and compact $Cr_2O_3$-based oxide layer on its surface when exposed to 25 MPa supercritical water at 625 °C for 500 h. This oxide layer acts as a protective barrier, preventing extensive oxidation of the substrate, which suggests its potential suitability for fuel cladding materials. Furthermore, Alloy 33 has demonstrated promising corrosion resistance in high-temperature steam environments up to 800 °C. As a result, Alloy 33 was chosen as a representative high chromium alloy for corrosion studies in simulated HTL environments. It should be noted that chromium- or nickel-based alloys are generally more expensive than austenitic stainless steels, as indicated in Table 5. However, our recent study [71] showed that high chromium-bearing nickel-based alloys exhibit excellent corrosion performance in simulated HTL conversion environments. Therefore, for long-term safe operation, it is advisable to consider the use of nickel-based or chromium-based alloys as suitable construction materials, despite their higher costs compared to austenitic stainless steels.

**Table 5.** Relative cost ratios of different types of steels and alloys [73].

| Types | Relative Cost Ratio Based on the Assumption of the Cost of SS304L = 1.0 |
|:---:|:---:|
| SS 310 | 2.75 |
| SS 316L | 1.25 |
| Alloy C-276 | 5.75 |
| Alloy 625 | 7.40 |
| Alloy 33 | 9.80 |

## 2. Experimental Procedure and Autoclave Testing Methodology

### 2.1. Testing Sample Preparation

The Alloy 33 plate used in this study was purchased from VDM Metals LLC., and its major chemical compositions are listed in Table 6, based on the provided mill test report. The other minor alloying elements, which are unlikely to significantly affect the corrosion performance, have been omitted in the table.

**Table 6.** Chemical composition of Alloy 33 used in this study.

| Alloy/Composition | Cr | Mo | Ni | Fe | Mn | C | Cu | Si | P | N |
|:---:|:---:|:---:|:---:|:---:|:---:|:---:|:---:|:---:|:---:|:---:|
| Alloy 33 | 32.8 | 1.49 | 30.7 | balance | 0.62 | 0.01 | 0.52 | 0.24 | 0.01 | 0.38 |

Testing coupons with rectangular shapes measuring 10 mm × 20 mm × 2 mm were machined from the plate of Alloy 33. Each coupon had a 4 mm diameter hole drilled near the top for mounting onto the Alloy 625 sample holder. Before conducting the autoclave test, every coupon underwent careful polishing and thorough cleaning, according to a standard procedure designed for high temperature corrosion testing [74]. More detailed information on sample preparation can be found in the references [17,37]. An identification (ID) was assigned to each sample and was marked on the upper-left corner to ensure accurate identification after HTL exposure. Prior to the corrosion tests, the freshly prepared

samples were weighed using a microbalance with a resolution of 1 μg and measured with a digital caliper of 0.001 cm precision. It is important to note that each autoclave corrosion test involved four prepared samples of Alloy 33 to improve the accuracy and precision of the corrosion results.

## 2.2. Autoclave Testing Methodology

The corrosion performances of Alloy 33 under simulated non-catalytic and catalytic HTL conversion conditions were evaluated through an autoclave test. The test matrix, outlining the specific conditions, is presented in Table 7. The tests were carried out in an Alloy C-276 autoclave, utilizing a pre-oxidized Alloy 625 liner (as shown in Figure 1), in accordance with ASTM G31, to minimize the influence of the autoclave wall alloy. The pre-oxidized Alloy 625 liner and Cr–Ni wires were oxidized at 400 °C for one day in an air furnace and were then used to hold the testing solutions and samples. The purpose of using pre-oxidized Cr–Ni wires was to prevent any galvanic effect between the testing samples and the sample holder. Once the samples and solution were placed inside the liner, the autoclave was sealed and purged with nitrogen gas for 60 min to create a de-aerated environment. A leak test was conducted under high-pressure $N_2$ conditions. The autoclave was then powered on to reach the desired temperature, and the tests were conducted for a duration of 10 days. After completing the test period, the autoclave heater was turned off, and the autoclave was allowed to cool down to room temperature. Subsequently, the exposed samples were carefully removed from the autoclave, cleaned using distilled water, dried with pressurized air, and reweighed using the same microbalance as before.

**Table 7.** Test matrix conducted to assess the corrosion performances of Alloy 33 under simulated non-catalytic and catalytic HTL conversion conditions.

| Test # | Testing Conditions |
|---|---|
| #1 | in static Alloy C-276 autoclave with an Alloy 625 liner containing de-aerated ultrapure water at 310 °C for 10 days |
| #2 | in static Alloy C-276 autoclave with an Alloy 625 liner containing de-aerated 0.5 M $K_2CO_3$ at 310 °C for 10 days |
| #3 | in static Alloy C-276 autoclave with an Alloy 625 liner containing de-aerated 0.5 M $K_2CO_3$ and 3500 ppm $Cl^-$ + 2500 ppm $S^{2-}$ at 310 °C for 10 days |
| #4 | in static Alloy C-276 autoclave with an Alloy 625 liner containing de-aerated ultrapure water at 365 °C for 10 days |
| #5 | in static Alloy C-276 autoclave with an Alloy 625 liner containing de-aerated 0.5 M $K_2CO_3$ at 365 °C for 10 days |

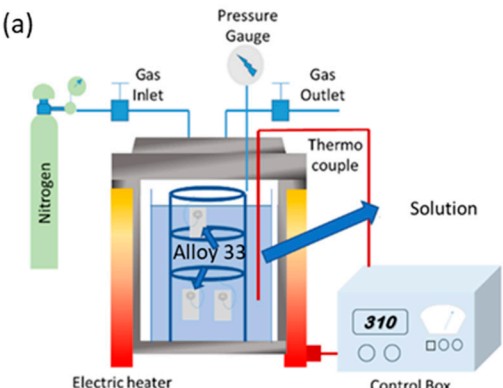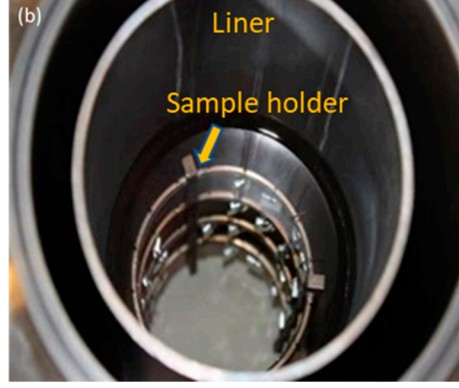

**Figure 1.** (**a**) Schematic of the static autoclave setup used in this study; (**b**) Top-view photograph of the Alloy 625 liner and sample holder.

### 2.3. Corrosion Rate Assessment

Two techniques are commonly employed to assess the corrosion rates of alloys in high-temperature aqueous solutions: direct mass change measurement, which determines the variation in mass of a sample before and after exposure, and indirect weight loss measurement, which involves removing the formed corrosion products from a sample (descaling) and then measuring the mass change before exposure and after descaling. Our recent studies [10,17] indicated that the indirect weight loss method is a more precise technique to assess corrosion rates in biomass conversion environments compared to direct mass change, particularly in the presence of catalysts, due to the following reasons:

- localized nodular oxidation and deposition of dissolvable corrosion products can occur on the alloys during autoclave cooling-down stage;
- the calculation of corrosion rates, by accounting for the molar mass ratio between the metal and metal oxides (weight gain method), is not accurate enough for assessing corrosion rates because the formed surface scales are not purely composed of a single oxide/hydroxide compound;
- the chemical dissolution rates of the formed oxide scales in ultrapure water can differ significantly from that in $0.5\,M\,K_2CO_3$, which further impacts the accuracy of the direct mass change method.

It is important to note that the direct mass change measurement assumes that surface oxidation of the alloy would be the dominant reaction during the high temperature exposure and that the formed corrosion products would be compact without localized nodular oxidation and/or spallation. Therefore, considering all these factors, weight loss serves as a more precise and accurate approach in assessing corrosion rates after exposure to catalytic HTL conversion environments.

The descaling process followed the same procedure described in detail in our previous studies [10,17]. The average corrosion rate of a sample ($\mu$m/year) was calculated using Equation (1) [75]:

$$Corr.\ Rate = \frac{8.76 \times 10^7 \bullet \Delta m}{dAt} \tag{1}$$

where $\Delta m$ (g) is the measured mass loss of a sample before exposure and after descaling, $A$ ($cm^2$) is the surface area of the coupon, $d$ ($g/cm^3$) represents the alloy density, and $t$ (h) represents the duration of exposure.

### 2.4. Corrosion Product Characterization

Following the corrosion test, the Alloy 33 sample with a direct mass change value closest to the average value in the set of four replicates was selected for dedicated microscopic characterizations using scanning electron microscopy (SEM), energy-dispersive X-ray spectroscopy (EDS), and Focus Ion Beam (FIB) techniques. Detailed information regarding the instruments used for the study and their operating procedures can be found in the references [10,17].

## 3. Results and Discussion

### 3.1. Effect of Catalyst on Corrosion Rate

As mentioned earlier, the corrosion rates of Alloy 33 were evaluated in simulated non-catalytic and catalytic HTL environments, and the results are presented in Figure 2. At 310 °C, the corrosion rate of Alloy 33 in pressurized hot water was approximately 1.8 $\mu$m/year, while the rate nearly doubled with the addition of 0.5 M $K_2CO_3$ catalyst. The disparity in corrosion rates was likely due to the accelerated dissolution rates of formed oxides at different pH values (around 5.7 in ultrapure water and approximately 11.2 in 0.5 M $K_2CO_3$ at 310 °C), which are discussed further in the following section. As the temperature increased to 365 °C, the corrosion rate of Alloy 33 in ultrapure water rose to approximately 2.3 $\mu$m/year. Our recent study [10] found that increasing temperature significantly increased the diffusion rates of ions, resulting in a higher overall oxide formation

rate. Considering that the chemical dissolution rate of Cr-enriched oxides (which were observed on Alloy 33 and are discussed in the following section) in high-temperature water occurs simultaneously, and is enhanced with temperature [76,77], the increased corrosion rate was likely due to the accelerated formation and dissolution rates of the formed oxides at the higher temperature. With the addition of 0.5 M $K_2CO_3$ catalyst at 365 °C, the corrosion rate further increased to approximately 4.2 µm/year, which could be attributed to the change in pH value and the enhanced oxide dissolution rate at a higher pH value (as described in Equation (2) in the next section). More importantly, the corrosion rates of the alloy were much lower than 100 µm/year at both temperatures, which is the suggested corrosion rate allowance for pressure vessels/reactors used in similar chemical plants over the projected service interval of 20 years [78,79]. This indicated that the alloy should be considered for the next phase of pilot-scale tests in real biomass HTL environments.

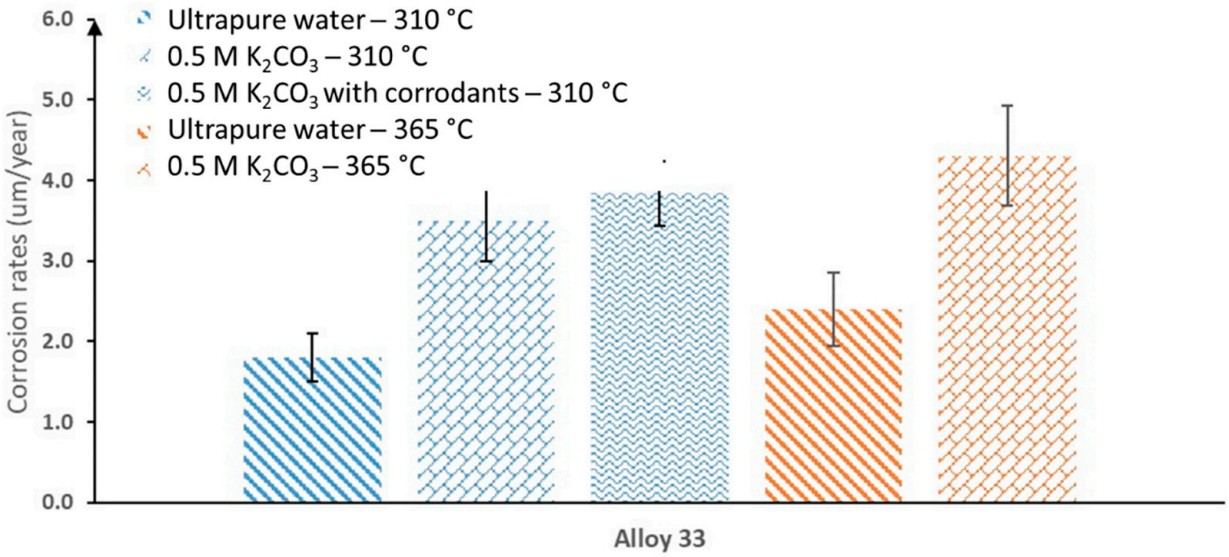

**Figure 2.** Comparison of the average corrosion rates of Alloy 33 in non-catalytic and catalytic HTL environments at 310 and 365 °C, respectively.

### 3.2. Characterization of Formed Oxide Layers

Top-view SEM images of as-polished and corroded Alloy 33 coupons exposed to ultrapure water and 0.5 M $K_2CO_3$ at 310 °C are shown in Figure 3. The oxide layer formed in ultrapure water was relatively thin with visible grinding lines on the surface, indicating minimal corrosion damage to the alloy in this environment. A limited number of particles were randomly distributed on the oxide layer. EDS spot analysis revealed that this layer primarily consisted of Cr and Fe oxides. However, when exposed to the catalytic environment, the oxide layer significantly increased in thickness, and nodular oxide particles were observed on the surface. The cross-sectional morphology of the corrosion layer, as depicted in Figure 4, underwent a complete transformation with the addition of 0.5M $K_2CO_3$. In the catalytic HTL environment, the corrosion layer consisted of an outer porous layer and an inner compact oxide layer. Large oxide particles were observed atop the oxide layer, likely formed through the re-deposition of soluble corrosion products during the autoclave cooling process.

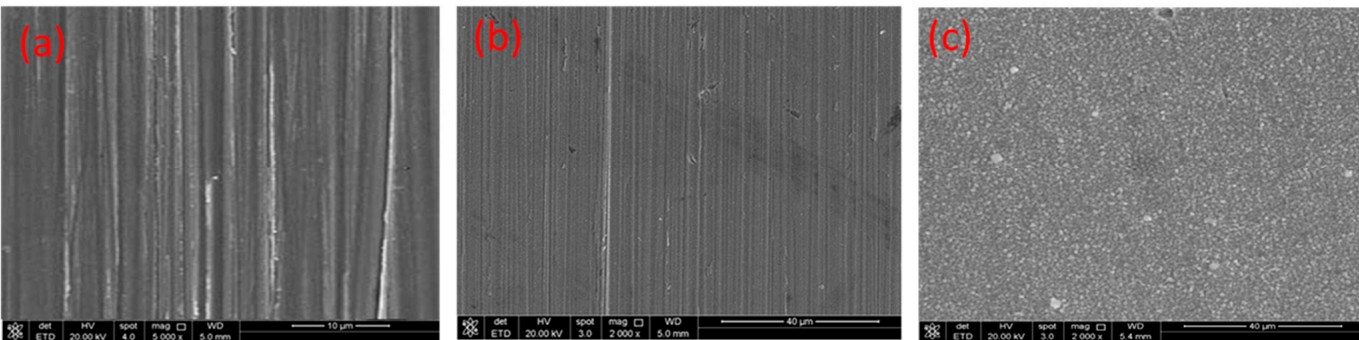

**Figure 3.** Top-view SEM images of Alloy 33 coupons: (**a**) as-polished, (**b**) after exposure to ultrapure water and (**c**) after exposure to 0.5 M $K_2CO_3$ solution at 310 °C.

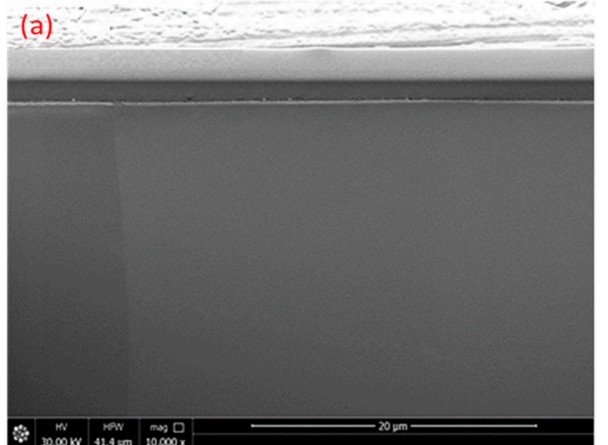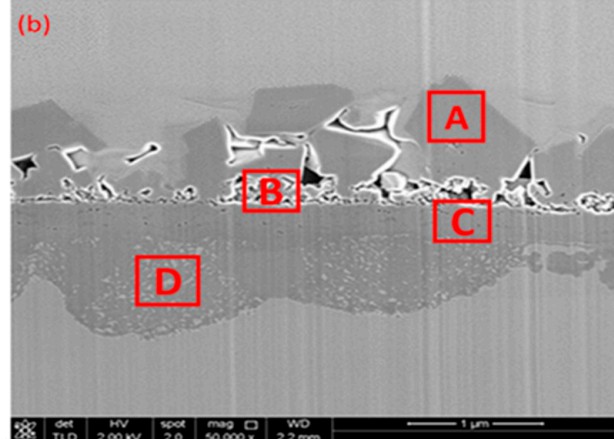

**Figure 4.** Cross-sectional SEM images of corroded Alloy 33 coupons after exposure to (**a**) ultrapure water and (**b**) 0.5 M $K_2CO_3$ solution at 310 °C, where area A represents oxide particles; area B represents outer oxide layer; area C represents inner oxide layer; and area D represents inner oxidation zone near the oxide/metal interface.

EDS analyses were conducted on the specific spots marked in Figure 4b to determine their compositions, and the results are presented in Figure 5. The large particles observed on the corrosion layer were primarily composed of iron oxides (spot A in Figure 4b), providing further evidence that they were formed through the re-deposition process. In contrast, the inner compact layer consisted predominantly of Cr-rich oxides (spot C in Figure 4b). It should be noted that nickel was also detected within the spot, likely in the form of cations, owing to the positive Gibbs free energy of formation of nickel oxides. Additionally, there was a thin layer composed of small oxide particles (spot B in Figure 4b). The EDS analyses indicated that these small particles could be a combination of iron oxides and chromium oxides, likely forming as nodular oxides at local defective sites. Based on our previous study [17], the presence of nickel hydroxides in this region was also plausible. Notably, relatively large oxide particles were observed in the inner layer near the oxide/metal interface (spot D in Figure 4b). Their formation might be attributed to the inward diffusion of oxygen along the grain boundaries. Further investigation is required to fully understand this observation.

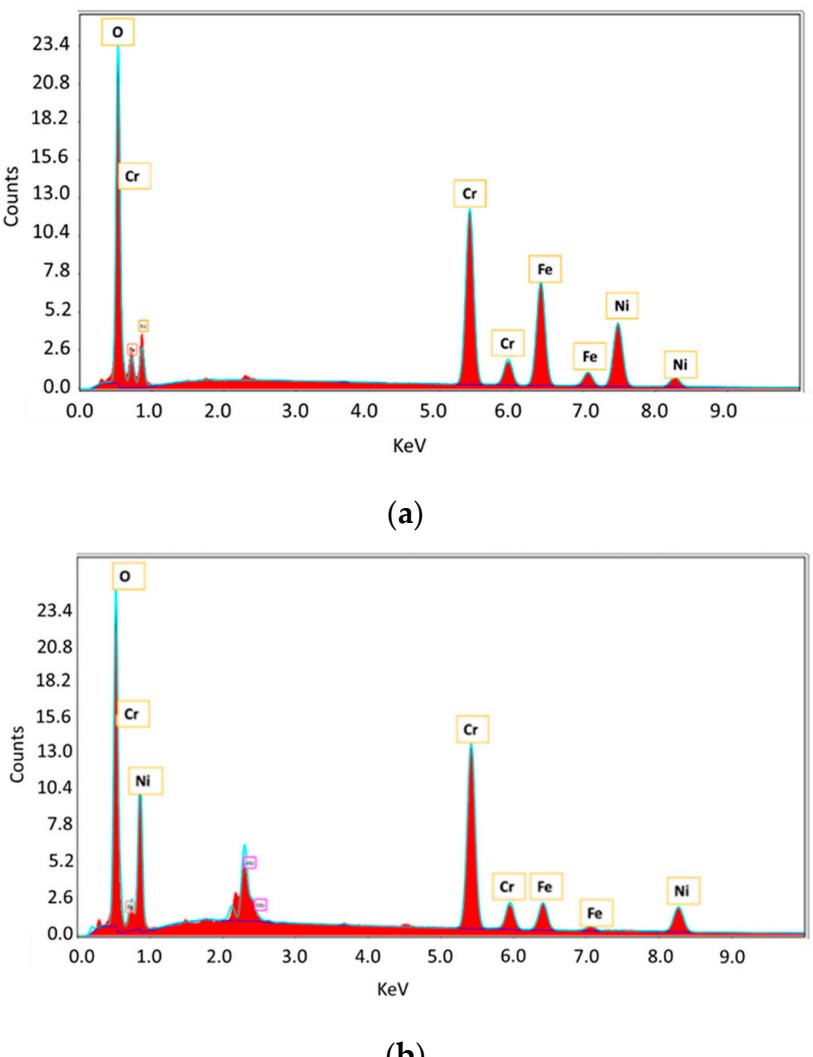

**Figure 5.** EDS spectrum collected on selected cross-sectional spots of Alloy 33 at (**a**) spot B; (**b**) spot C after exposure to 0.5 M $K_2CO_3$ solution at 310 °C.

Thermodynamic calculations conducted in our previous study [17] indicated that the Fe, Cr or Fr/Cr oxides could form and be stable in pressurized hot water (pH~5.7 at 310 °C), but became unstable in a catalytic HTL environment (pH~11.2). In the catalytic HTL environment, the chemical dissolution of these oxides would occur via the following hydrolysis reaction:

$$M_2O_3 + 2\,OH^- + 3\,H_2O = 2\,M(OH)_4^- \tag{2}$$

where M denotes Fe or Cr. In addition to chemical dissolution, the presence of a catalyst in the HTL environment may also enhance the rate of oxide growth, as indicated by the oxide layer thickness, shown in Figure 4. This observation can be attributed to the following factors. Firstly, with the addition of the catalyst, more defects could form in the corrosion layer, acting as diffusion pathways for cations/anions during oxide growth. Secondly, the presence of the catalyst might enhance the inward diffusion of $O^{2-}$, as suggested by Figure 4b. Previous data and calculations also indicated that the diffusion rates of oxygen anions in iron oxides and chromium oxides are significantly lower than those of Fe or Cr cations [80,81]. This suggests that the inward diffusion of oxygen is likely the rate-limiting factor during the growth of Fe oxides and chromium-enriched oxides, and $O^{2-}$ inward diffusion within the oxide is likely to be enhanced with the introduction of $K_2CO_3$ catalyst [10,17].

The formed oxide layer and corrosion rates of Alloy 33 were compared to Fe-based steels with chromium contents ranging from 9% to 22% [71]. It was found that Alloy 33 exhibited much better corrosion resistance in both non-catalytic and catalytic HTL environments. Notably, the significant difference in corrosion rates was likely attributed to the distinct composition of the oxide layer. Previous studies have shown that steels with chromium content less than 22 wt.% generally form oxide layers primarily composed of $Fe_3O_4$ and $Fe_{3-x}Cr_xO_4$ when exposed to high-temperature aqueous solutions [71,82,83]. In contrast, a compact Cr-enriched inner oxide layer is formed on Alloy 33. Considering the fact that inward oxygen diffusion is the rate-limiting step and considering the application of the point defect model (PDM) in similar corrosion systems [84], the corrosion of Alloy 33 in catalytic HTL environments is likely diffusion-controlled due to the relatively compact $Cr_2O_3$ layer. On the other hand, the corrosion of Fe-based steels with chromium content < 22 wt.% is likely controlled by the dissolution rates of the formed oxides on the surface. Moreover, when comparing the corrosion rates of Alloy 33 and Alloy 625 [71], it was observed that Alloy 33 exhibited only slightly better corrosion resistance in both non-catalytic and catalytic environments. This suggests that further increasing the chromium content (when the chromium content is already above 22%) would unlikely lead to a significant improvement in corrosion resistance during HTL conversion.

## 4. Conclusions

During the HTL conversion, biomass feedstock types, temperature, pressure, and the homogeneous and heterogeneous catalysts are important operating parameters affecting carbon conversion efficiency and the properties of the produced bio-oil, and their roles were overviewed in this study. From the corrosion perspective, temperature, catalyst, and inorganic corrodants generated during the conversion process are major factors that influence the corrosion performance of the conversion reactor and other core component alloys for safe operation. The study also investigated their impacts on the corrosion of a representative construction alloy (Alloy 33). It was observed that increasing temperature, the introduction of $0.5\,M\,K_2CO_3$ catalyst, and the presence of trace amounts of corrodants ($S^{2-}$ and $Cl^-$) led to a noticeable increase in the corrosion rate. Among these factors, the alkaline catalyst emerged as the dominant influence, as its addition completely changed the morphology and chemistry of the surface oxide scales grown on the alloys. Based on design requirements, Alloy 33 is considered a suitable candidate for further pilot-scale validation assessment in real biomass-containing HTL environments.

**Author Contributions:** Conceptualization, M.L. and Y.Z.; methodology, M.L. and Y.Z.; validation, M.L. and Y.Z.; formal analysis, M.L. and Y.Z.; investigation, M.L. and Y.Z.; resources, M.L. and Y.Z.; data curation, M.L. and Y.Z.; writing—original draft preparation, M.L.; writing—review and editing, Y.Z.; visualization, M.L. and Y.Z.; supervision, Y.Z.; project administration, Y.Z.; funding acquisition, Y.Z. All authors have read and agreed to the published version of the manuscript.

**Funding:** This research was funded by the Canadian NRCan OERD Clean Energy and Forest Innovation programs.

**Institutional Review Board Statement:** Not applicable.

**Informed Consent Statement:** Not applicable.

**Data Availability Statement:** Data available on request due to technical and time limitations. Concurrently, the data also forms part of an ongoing study.

**Acknowledgments:** The authors gratefully acknowledge the contribution and technical support of the staff (Jian Li, Pei Liu, Magdalene Matchim, Chao Shi) at CMAT (CanmetMATERIALS) corrosion and advanced microscopy laboratories.

**Conflicts of Interest:** The authors declare no conflict of interest. The funders had no role in the design of the study; in the collection, analyses, or interpretation of data; in the writing of the manuscript; or in the decision to publish the results.

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
