# Peer review of "Key Processing Factors in Hydrothermal Liquefaction and Their Impacts on Corrosion of Reactor Alloys"

_sustainability, doi:10.3390/su15129317_

Round 1

Reviewer 1 Report

1. Please insert more references after 2019 because most of them are before 2019. An overview/review requires to include the latest advancement in technology and materials.

2. Is this a review manuscript or result manuscript, please reflect it properly on the title.

3. Table 5 and 6 require citation or references to support.

Reviewer 2 Report

The article ‘’Overview of Hydrothermal Liquefaction Process and Corrosion Investigation on Reactor Materials for industrial deployment’’ is interested article, however, it require major changes. Moreover the technical, novel side of the paper is very week. Following are the comments for the authors to improve the article:

Ø  The article title is a like statement, authors are suggested to re-write the title with the research impact and novelty perspective.

Ø  Avoid using so many words in the title.

Ø  A lot of research is going on B Hydrothermal Liquefaction Process. How authors claim the new aspect for the need of this publication?

Ø  Problem statement should be mentioned at the start of the abstract, it is totally missing.

Ø  There should be some statistical figured values in the abstract which can quantify the research / optimization and it can make readership of the journal easy.

Ø  Abstract needs to re-write as it is not clear and number of abstract components are missing.

Ø  There should be some proper synchronization of the sentences in meaningful way.

Ø  The abstract should also include the solution of the problem based on the problem statement with some particular application/s.

Ø  Key words include abbreviation. Authors should add at least 5 keywords and also specific meaningful keywords.

Ø  Formatting and style of all tables must be same.

Ø  Heading 2 should be bold, however, it is suggested to merge this in introduction section.

Ø  Cumulative references should be avoided at multiple places i.e. page 2 [3-5], [9-11].

Ø  Authors should mention references in Table 1 and 2 like in table 3.

Ø  Axes on figure 5 are not visible.

Ø  The quality of figures is very poor. All figures require re-work.

Ø  There are few old references, authors are encouraged to add latest literature.

Ø  The graphics of the results need improvement. It should be reviewed by the authors in the revised version.

Ø  Authors should only focus on the conclusion of the research and not to add results and/or any other information.

Reviewer 4 Report

The manuscript "Overview of Hydrothermal Liquefaction Process and Corrosion Investigation on Reactor Materials for industrial deployment" presented important results about corrosion on reactor used in HTL process. It is well written and the results are promising for advance in construction of  industrial reactors.

Only two points deserve some considereation:

1- The authors wrote HHV a couple of times before they give the real mean of HHV (higher heating value), I recommend to correct this;

2- For a better comparison it is recommended to show the EDS and SEM images of original Alloy 33, before the corrosion test.

Round 2

Reviewer 1 Report

All comments have been addressed

Reviewer 2 Report

accept

Reviewer 3 Report

THE REQUESTED MODIFICATIONS HAVE BEEN DONE CORRECTLY.